# Prolonged Fasting Induces Histological and Ultrastructural Changes in the Intestinal Mucosa That May Reduce Absorption and Revert after Enteral Refeeding

**DOI:** 10.3390/nu16010128

**Published:** 2023-12-30

**Authors:** Gonçalo Nunes, Marta Guimarães, Hélder Coelho, Ricardo Carregosa, Cátia Oliveira, Sofia S. Pereira, António Alves de Matos, Jorge Fonseca

**Affiliations:** 1Gastroenterology Department, GENE—Artificial Feeding Team, Hospital Garcia de Orta, 2805-267 Almada, Portugal; 2ICBAS-UP—Instituto de Ciências Biomédicas Abel Salazar, Universidade do Porto, 4050-313 Porto, Portugal; 3PaMNEC—Grupo de Patologia Médica, Nutrição e Estudos Clínicos, Egas Moniz Center for Interdisciplinary Research (CiiEM), Egas Moniz School of Health and Science, 2829-511 Almada, Portugal; 4UMIB—Unit for Multidisciplinary Research in Biomedicine, ICBAS—School of Medicine and Biomedical Sciences, Universidade do Porto, Rua Jorge Viterbo Ferreira 228, 4050-313 Porto, Portugal; 5ITR—Laboratory of Integrative and Translocation Research in Population Health, Rua das Taipas 135, 4050-600 Porto, Portugal; 6Pathology Department, Hospital Garcia de Orta, 2805-267 Almada, Portugal; 7Cmicros—Centro de Microscopia Eletrónica e Histopatologia, CiiEM—Centro de Investigação Interdisciplinar Egas Moniz, 2829-511 Almada, Portugal

**Keywords:** malnutrition, PEG, duodenal mucosa, histology, ultrastructure

## Abstract

**Background:** Malnutrition is usual in patients referred for endoscopic gastrostomy (PEG). Refeeding syndrome is rarely observed in PEG-fed patients, which could possibly be associated with reduced absorption induced by prolonged starvation. **Objective:** In patients submitted to PEG after a significant period of fasting, the present study aims to: 1. evaluate the histological/ultrastructural initial changes in the intestinal mucosa, potentially associated with reduced absorption, and 2. assess if these changes could reverse with enteral refeeding. **Methods:** The present study is an observational, prospective, controlled study. Adult patients with ingestion below 50% of daily needs for at least one month and/or diagnosis of malnutrition were enrolled. Duodenal biopsies were taken at baseline and after 3–6 months of PEG feeding, which then underwent histological/ultrastructural analysis. Random healthy individuals were used as controls. **Results:** A total of 30 patients (16 men/14 women) aged 67.1 ± 13.5 years were included. Malnutrition was found in 40% of patients. Approximately 14 patients completed follow-up during both periods (46.7%). At baseline: duodenal mucosal atrophy was evident in three patients (10%); the median villi length (MVL) was 0.4 mm (0.25–0.6 mm), with it being shorter than the controls, which was 0.6 mm (0.4–0.7 mm) (*p* = 0.006); ultrastructural changes included focal shortening, bending, and disruption of enterocyte microvilli, the presence of citoplasmatic autophagic vacuoles, dilation and vesiculation of the smooth endoplasmic reticulum, and the presence of dilated intercellular spaces with basement membrane detachment. After refeeding, most patients displayed normal histology (92.9%) and increase MVL (*p* < 0.001), ultrastructural changes disappeared, and enterocytes resumed a normal appearance, although retaining scarce, small, dense bodies in apical regions from the evolution of previous autophagy. **Conclusions:** Prolonged fasting induces histological and ultrastructural changes in the intestinal mucosa that may reflect impaired absorption in the early post-PEG period. These changes were reverted after refeeding with enteral nutrition.

## 1. Introduction

Clinical nutrition and especially artificial nutrition are major areas of gastroenterology clinical practice. Several gastrointestinal disorders interfere with digestive function, induce systemic inflammation, and influence physiologic metabolic pathways, which may affect energy and protein balance [1].

Refeeding syndrome (RS) is a life-threatening condition defined as potentially fatal shifts in fluids and electrolytes, resulting from metabolic and hormonal changes that may occur in malnourished patients receiving nutritional therapy. The main clinical features of RS include fluid balance changes, abnormal glucose metabolism, hypophosphatemia, hypomagnesemia, hypokalemia, and vitamin deficiency, leading to cardiorespiratory failure, malignant dysrhythmias, neuromuscular symptoms, hematologic dysfunction, and death [2,3,4]. Although no strict internationally agreed definition has been established, the criteria of the *National Institute for Health and Clinical Excellence* (NICE) were developed to identify high RS risk. Most gastroenterological patients eligible for artificial nutrition fulfil the NICE criteria; nevertheless, RS prevalence and its real contribution to clinical outcomes has not been explored in the current literature. Close monitoring and prevention should be the priority, which includes starting nutritional support at a low level of energy replacement, vitamin supplementation, and the correction of all electrolytes and fluid imbalances [2,4,5,6,7,8].

Percutaneous endoscopic gastrostomy (PEG) is a safe and well-tolerated method for long-term enteral feeding, with it being recommended when a period of inadequate oral intake exceeding 3–4 weeks is expected [9,10].

Patients undergoing PEG due to chronic disease with prolonged dysphagia are a particular subgroup that could be at high risk of RS since ingestion below 50% of daily needs is usually observed for several weeks and established malnutrition is highly frequent before referral to the artificial nutrition team. As current practice of our artificial feeding team, most PEGs are performed in an outpatient regimen, with patients being discharged a few hours after the procedure [11,12]. In this clinical setting, patients and their caregivers are empirically instructed to provide reduced enteral food volumes during the first days after the procedure in order to avoid RS. However, the recommendations of starting PEG feeding with hypocaloric regimens might not always be accomplished, perhaps due to caregivers’ fear of maintaining malnutrition. Even so, clinical and laboratory signs of RS are seldom identified in PEG patients [13]. A possible explanation for this fact could be the loss of absorptive capacity due to intestinal mucosal atrophy induced by severe malnutrition, which may prevent complete nutrient digestion/absorption in the early post-PEG period. Although some animal model studies have already demonstrated that long periods of fasting may induce morphological changes in the intestinal mucosa, this has never been evaluated in human studies, and the impact of refeeding after enteral nutrition in reverting these histopathological changes is unknown [14,15,16,17,18,19]. In the clinical setting of PEG patients, the confirmation of this hypothesis could contribute to optimizing nutritional support, allowing for the optimization of early post-PEG feed nutritional recovery.

The present study aims to:Analyze the histological and ultrastructural changes in the intestinal mucosa of malnourished patients referred and submitted to endoscopic gastrostomy after a significant fasting period.Assess if these changes are reverted after refeeding through enteral nutrition by PEG.

## 2. Material and Methods

### 2.1. Study Design

This single-center, observational, prospective study was conducted in a tertiary hospital. The present study was approved by the institutional ethics committee, under authorization 52/2016.

### 2.2. Patients

Consecutive ambulatory patients aged 18 years old or older, referred for PEG with the Artificial Feeding Team (GENE) of the Gastroenterology Department of Hospital Garcia de Orta, were initially eligible. Additional inclusion criteria included oral ingestion below 50% of energy daily needs for a minimum period of one month and/or baseline diagnosis of malnutrition according to the European Society of Clinical Nutrition and Metabolism (ESPEN) criteria [20], and acceptance to participate in the study and follow-up. For each patient enrolled, the following methodology was applied:Nutritional anamnesis performed by a trained dietitian through a dietary recall.Estimation of energy needs according to age, gender, height, weight, and clinical setting to assess the reduction in food ingestion in the last month before PEG.Prospective clinical and demographic database registry that includes age, gender, clinical indication for PEG, anthropometry, laboratory parameters, and survival.PEG was performed by two gastroenterologists using the “pull” method described by *Gauderer-Ponsky*, with the patients under deep sedation. Antithrombotic therapy was managed according to international guidelines [21].The collection of 4–6 duodenal biopsies (second portion) at the time of the gastrostomy procedure (T0) and after 3–6 months of PEG feeding at endoscopic tube replacement (T1).

All participants or their representatives signed an informed consent form. Patients with oncologic disease, patients who quantified food ingestion above 50% of energy daily needs during the last month before PEG, those with evident inability to provide credible anamnestic data, or those who refused to participate in the study were excluded. The technical inability to obtain duodenal biopsies and the expected impossibility of having an appropriate clinical and nutritional follow-up were additional exclusion criteria.

Random healthy individuals who had undergone upper gastrointestinal endoscopy to investigate ferropenic anemia also had duodenal biopsies taken, with them being used as baseline negative controls if no endoscopic and histologic abnormality was diagnosed and a non-duodenal source of bleeding was found.

### 2.3. Anthropometric Evaluation

The anthropometric evaluation was performed by a dietitian before the procedure, according to the ISAK manual of the International Society for the Advancement of Kinanthropometry [22]. The average of three consecutive measurements was recorded on the patient’s clinical file.

Body mass index (BMI) was determined in most patients using the equation weight/height^2^, measured in kilograms and meters, respectively. If patients were unable to easily stand up for weight and height evaluation, BMI was estimated using the mid-upper arm circumference (MUAC) and the regression equations described by Powell-Tuck and Hennessy [23], which have been previously used and proven to provide a reliable BMI estimation in PEG-fed patients [24].

MUAC was measured in centimeters, using a flexible measuring tape wrapped around the mid-upper arm, halfway between the olecranon and the acromion process.

Patients were classified as having malnutrition if they experienced unintentional weight loss above 10% indefinite of time or above 5% over the last 3 months plus a BMI below 20 Kg/m^2^ if under 70 years of age or a BMI below 22 Kg/m^2^ if 70 years old or older.

### 2.4. Laboratory Evaluation

A blood sample was obtained just before the PEG procedure. Serum albumin, transferrin, total cholesterol, and electrolytes, namely phosphorus, magnesium, and potassium, were measured at baseline (T0) and during follow-up (T1) as part of patient global nutritional and metabolic evaluation. Values of albumin below 3.5 g/dL, transferrin below 200 mg/dL, and total cholesterol below 160 mg/dL were considered low values, suggestive of malnutrition and/or poor prognosis. Normal cut-off values for serum electrolytes were considered, according to the laboratory institutional protocol, namely: phosphorus: 2.5–4.8 mg/dL, magnesium: 1.5–2.1 mg/dL, and potassium: 3.5–5.0 mmol/L. Patients with low levels of these serum electrolytes did not start enteral nutrition refeeding until replacement and complete normalization.

### 2.5. Duodenal Biopsy Analysis

The duodenal biopsies were analyzed for histological and ultrastructural changes in both study moments. Histologic analysis was performed using optical microscopy with hematoxylin–eosin staining observed by two pathologists, including one expert gastrointestinal pathologist. All samples were analyzed for the presence of mucosal atrophy according to the adapted Marsh–Oberhuber classification and average duodenal villi length in micrometers, with at least two villi being measured through the use of an automatic scale.

For transmission electron microscopy, fragments from the duodenal biopsies were fixed in 3% glutaraldehyde in cacodylate buffer (0.1 M, pH 7.4) overnight at 4 °C. The fixed fragments were rinsed and post-fixed in 1.0% osmium tetroxide for 1 h in the same buffer and dehydrated in graded ethanol passages. The fragments were embedded in a mixture of Epon and Araldite after two 15-min passages in propylene oxide. Thin sections, cut on a Reichert Ultracut II Ultramicrotome with a diamond knife, were stained on the grid with uranyl acetate and lead citrate and examined with a JEOL 1200EX electron microscope.

All patient histologic and ultrastructural changes were compared with the duodenal samples of healthy controls.

### 2.6. Statistical Analysis

The statistical analysis was performed using the Statistical Package for Social Sciences (SPSS^®^ Inc., Chicago, IL, USA) and Microsoft Office Excel Professional 2017^®^. Normality was assessed using the Kolmogorov–Smirnov test. Continuous variables were expressed as the mean ± standard deviation and categorical variables were expressed as percentages. Student’s t-test or the Mann–Whitney test was used for continuous variables. All tests were performed at a 5% level of statistical significance.

## 3. Results

### 3.1. Patients

Thirty patients who fulfilled the inclusion criteria were initially included in the study: 16 men and 14 women, aged between 38 and 87 years (mean 67.1 ± 13.5). Thirteen patients (43.3%) were below 70 years of age. The main characteristics of the population are described in Table 1. The clinical indication for PEG was prolonged dysphagia due to amyotrophic lateral sclerosis (90%), post-stroke (6.7%), and esophageal motility disturbance (3.3%). All patients were eligible for PEG feeding without previous tube feeding due to chronic disease with anticipated prolonged dysphagia. From 30 included patients, 14 completed the protocol at the two time periods (46.7%), 9 died before the end of the study (30%), 1 removed the PEG tube due to dysphagia resolution (3.3%), and 6 were lost to follow-up (20%). From the patients who died during follow-up, the median survival was 5.5 months. There were no major post-PEG adverse events, and no death was attributable to a complication of the technique. All deaths were caused by the evolution of the underlying disorder.

A total of 10 patients (3 men and 7 women) aged between 42 and 77 years (mean 52.7 ± 11.3) were included in the control group undergoing upper gastrointestinal endoscopy with duodenal biopsies to investigate ferropenic anemia and rule out malabsorption. All of them presented normal endoscopic evaluation, and no cause for anemia was detected in the upper gastrointestinal tract.

The patient flowchart is described in Figure 1. Patient flowchart: 30 patients were enrolled in the study and underwent PEG for long-term enteral nutrition after clinical, anthropometric, and laboratory evaluation. Duodenal biopsies were taken during the procedure for histologic and ultrastructure analysis. In total, 14 patients completed the study protocol, repeating duodenal mucosa assessment after 3–6 months of enteral refeeding. Normal healthy individuals submitted to upper GI endoscopy with duodenal biopsies were used as negative controls.

### 3.2. Nutritional Assessment: NRS 2002, Anthropometry, and Laboratory Data

All patients presented with a major reduction in oral ingestion and significant weight loss. NRS 2002 was ≥3 points in all patients, signaling high nutritional risk.

BMI before PEG ranged from 14.5 to 27.9 Kg/m^2^ (mean 21.56 ± 3.5 Kg/m^2^) and was considered low in 12 patients (40%) after being adjusted to the age group. On the day of the gastrostomy, the mean albumin was 4.1 ± 0.5 g/dL, transferrin was 220.2 ± 42 mg/dL and total cholesterol was 173 ± 37 mg/dL, being this being considered low in 1 (3.3%), 9 (30%), and 8 (26.7%) patients, respectively. Although albumin, transferrin, and total cholesterol are dependent on several factors, low serum levels together with reduced ingestion, weight loss, and low BMI inferred a significant prevalence of malnutrition at baseline (N = 12; 40%). Serum electrolytes were normal at the time of PEG in all patients. BMI and laboratory parameters did not change significantly after refeeding (*p* > 0.05).

Nutritional assessment data are highlighted in Table 1.

### 3.3. Duodenal Mucosa Histology

At baseline (T0), 27 patients displayed normal duodenal mucosa with no criteria of atrophy (90%). Atrophy was observed in three patients (10%), classified as mild (Marsh 3A) in two patients and moderate (Marsh 3B) in one patient. Conversely, 22 patients (73.3%) presented villi below 0.5 mm at T0. In the control group, all patients displayed normal histology without atrophy; nevertheless, villi length below 0.5 mm was present in 30% of the controls. The median villi length was 0.4 mm (0.25–0.6 mm) in the patients and 0.6 mm (0.4–0.7 mm) in the control group with a significant difference (*p* = 0.006). No association was observed between baseline villi length and patient survival (*p* > 0.05).

After refeeding, duodenal histology could be assessed in 14 patients. Most patients displayed normal histology (N = 13, 92.9%), and mild atrophy (Marsh 3A) was present in only one patient (7.1%), with a median villi length of 0.5 mm (0.35–0.85). The median villi length significantly increased between the two periods (*p* < 0.001).

Table 2 describes the optical microscopy analysis and follow-up of all included patients. Figure 2 and Figure 3 exemplify the duodenal villi morphology in both time periods.

### 3.4. Duodenal Mucosa Ultrastructure

The enterocytes of all included patients who were submitted to prolonged fasting (T0) showed several degenerative changes (Figure 4). The microvilli of the striated plate showed irregularities consisting of focal shortening, bending, and disruption of the regular arrangement, which may compromise the barrier and absorptive function of this structure. The apical region of the enterocytes disclosed autophagic vacuoles and dense bodies probably corresponding to primary lysosomes. This represents autophagic phenomena associated with prolonged fasting. Many enterocytes showed dilatation and vesiculation of the smooth endoplasmic reticulum. The basal zone of the epithelium frequently showed dilated intercellular spaces, often filled with a medium-density granular material. In some cases, the basal cell processes were detached from the underlying basement membrane. These alterations indicate a significant compromise of the normal cell physiology on the gut epithelial lining and disruption of the permeability barrier. However, no evident alterations of the junctional complexes could be found. All of these changes were not observed in the controls, who displayed normal cell architecture with regular microvilli structure and normal organelles (Figure 5).

After refeeding through enteral nutrition, most of the described alterations disappeared and the cells regained a normal appearance, although retaining dense bodies in the apical region, probably representing residual lysosomes resulting from the evolution of the autophagic vacuoles (Figure 6).

## 4. Discussion

Protein–energy malnutrition is a highly prevalent condition in gastroenterology clinical practice, especially when food intake is severely reduced for long periods. It is associated with poor outcomes in ambulatory and hospitalized patients, causes prolonged hospital stays, increases infectious and perioperative complications, leads to higher healthcare costs, and increases morbidity and mortality [25].

Enteral nutrition through PEG is an effective approach to manage malnutrition when oral feeding is compromised. Several clinical conditions have become common indications for PEG feeding, mostly in cases of long-term dysphagia due to oncologic disease (mainly head and neck or esophageal cancer) and neurologic disorders like amyotrophic lateral sclerosis (ALS), cerebrovascular disorders, dementia, central nervous system trauma, and congenital or neonatal abnormalities [25]. Previous studies have reported that more than 50% of patients presented a low BMI and reduced concentration of serum proteins at the time of PEG, signaling a high prevalence of malnutrition associated with systemic inflammation [13].

In the current study, the authors analyzed 30 patients referred for PEG with significantly low caloric intake and absence of previous nutritional support via tube feeding, looking for histologic and ultrastructural changes in the intestinal mucosa. A large number of patients were diagnosed with ALS, as a significant reduction in oral ingestion is commonly observed from the early stages of this neurodegenerative disease and our team is a reference center for enteral feeding in ALS patients. During the course of ALS, a decline in nutritional status is common and malnutrition is an independent, negative, prognostic indicator for survival. Weight loss may be explained by several factors, including reduced food intake associated with decreased strength for food manipulation and chewing, impaired salivary secretion, swallowing difficulties, constipation, and fear of pulmonary aspiration. ALS patients have also been associated with significant gastrointestinal dysfunction that contributes to malnutrition as the enteric nervous system may become affected, motor neuron disorders develop, and dysbiosis may occur [26]. Cancer patients usually present significant states of systemic inflammation and are subjected to cytotoxic therapies that themselves change intestinal homeostasis; therefore, their inclusion was avoided. Also, we favored the recruitment of patients who maintained intact cognitive functions, including several ALS patients, in order to prevent some ethical issues regarding alternative inclusion of patients with dementia, post-stroke persistent dysphagia, neurotrauma, and other conditions that lead to severe mental dependence. The prevalence of malnutrition reached almost 40% of the sample, which was consistent with the literature. Nevertheless, most laboratory parameters were normal, with less than 30% of patients presenting low serum protein and total cholesterol levels, which could be explained by the reduced contribution of systemic inflammation to the development of malnutrition in the early stages of ALS.

The histopathologic evaluation of duodenal biopsies showed a low prevalence of mucosal atrophy (10%) according to the standard analysis described by the Marsh–Oberhuber classification. However, the hypothesis of impaired gut function with the loss of absorption induced by prolonged fasting may manifest in more subtle changes like reduced enterocyte mass, intestinal inflammation, and increased bacterial translocation, not commonly easily assessed in clinical practice [27]. Even in the scarcity of traditional criteria for mucosal atrophy, the authors hypothesized that median villi length could be a surrogate marker of subtle duodenal dysfunction that may lead to impaired absorption. The normal villi length in humans is not clearly established, with old studies reporting expected variation between 0.4 mm to 1 mm [28,29]. In the present study, more than two-thirds of patients presented a median villi length below 0.5 mm, which was significantly shorter compared to the controls. Almost all patients who completed follow-up displayed normal histology, median villi length significantly improved between the two study periods, and all patients who fulfilled the criteria of mucosal atrophy improved after refeeding. These results suggest that fasting is associated with morphologic changes in duodenal histology manifested as decreased villi length and mucosal atrophy, which can be reverted through the normalization of energy intake. The degree of mucosa damage may be related to the fasting period and severity and can be linked to absorptive dysfunction.

The ultrastructural analysis of the duodenal mucosa also revealed changes at the cellular level. After a significant period of reduced energy intake, enterocytes displayed changes in the striated plate with disarrangement of microvilli. In these cells, increased autophagy was evident, demonstrated by the presence of cytoplasmic vacuoles. Dense bodies and vesiculation of the endoplasmic reticulum were also present. Beyond the evidence of enterocyte dysfunction, an increased length of intercellular spaces and the cell detachment of basal membranes may also contribute to compromised absorption. All of these changes were not observed in the controls and seemed to revert after 3 to 6 months of enteral nutrition with adequate intake. These findings are innovative and reflect metabolic derangement and impaired enterocyte absorptive capacity. As no previous study has evaluated the impact of prolonged fasting in intestinal mucosa using villi length measurement or electron microscopy, these findings represent a major advance in the comprehension of fasting-induced intestinal dysfunction.

The present study also displays some minor limitations. First, 30 patients were initially included and assessed at baseline; however, only 14 completed the follow-up, and the duodenal biopsies could be reanalyzed after refeeding—this occurred mostly because one-third died before endoscopic control due to disease evolution, one patient recovered from dysphagia, and six were lost to follow-up. Second, the histologic and ultrastructural analysis was performed in patients in two different periods, and the findings were compared to healthy controls; however, this evaluation was not blinded. Nevertheless, the biopsies were analyzed by at least two operators using the same standardized protocols to avoid inter-observer variability. Finally, the association of the structural anomalies found in the duodenal mucosa and a potential decrease in enterocyte absorptive capacity is indirect as no “functional” analysis was performed to assess it. In fact, there is no gold standard method to study absorption in this clinical setting and functional tests are not widely available in our country. An immunohistochemistry-based analysis of the intestinal mucosa to study the expression of digestive hormones is also planned by our group to overcome this bias and reinforce the obtained results. Metabolomics may also help to assess the impact of fasting and subsequent refeeding in absorptive mechanisms. Further studies with a similar patient population are planned, aiming to define possible serum biomarkers involved in refeeding syndrome and its impact on nutritional and clinical prognosis.

## 5. Conclusions

In conclusion, the results obtained in the current study support the authors’ hypothesis that prolonged fasting induces histological and ultrastructural changes that reflect decreased absorption. This includes shortening of the villi with or without mucosal atrophy resulting from degenerative changes in individual enterocytes. This impaired absorption may be responsible for a low prevalence of refeeding syndrome observed in the early post-PEG period. As the present study proves, after an immediate period of enteral refeeding, enteral nutrition not only corrects malnutrition but also normalizes histologic and ultrastructural changes in the duodenal mucosa—re-establishing absorption and digestive homeostasis.

Those findings have significant implications for clinical practice since they call for a wide discussion on the necessity of starting nutritional support using severe hypocaloric regimens in nutritional risk patients due to fear of RS. A less restrictive approach could enable nutritional recovery and improve clinical outcome, and this hypothesis may justify further studies.

## Figures and Tables

**Figure 1 nutrients-16-00128-f001:**
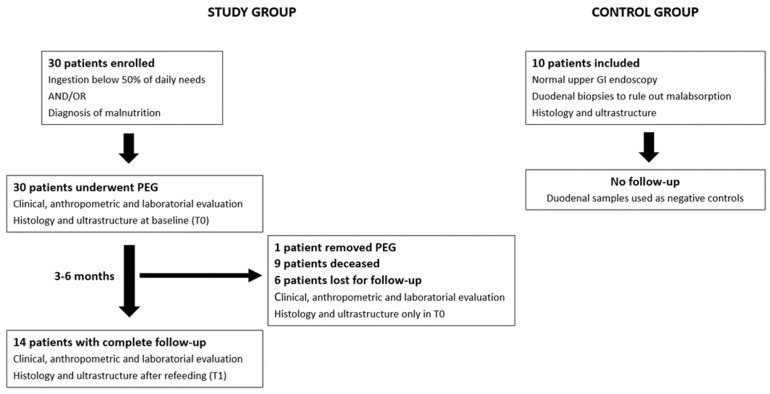
Patient flowchart: 30 patients were enrolled in the study and underwent PEG for long-term enteral nutrition after clinical, anthropometric, and laboratory evaluation. Duodenal biopsies were taken during the procedure for histologic and ultrastructure analysis. In total, 14 patients completed the study protocol, repeating duodenal mucosa assessment after 3–6 months of enteral refeeding. Normal healthy individuals submitted to upper GI endoscopy with duodenal biopsies were used as negative controls.

**Figure 2 nutrients-16-00128-f002:**
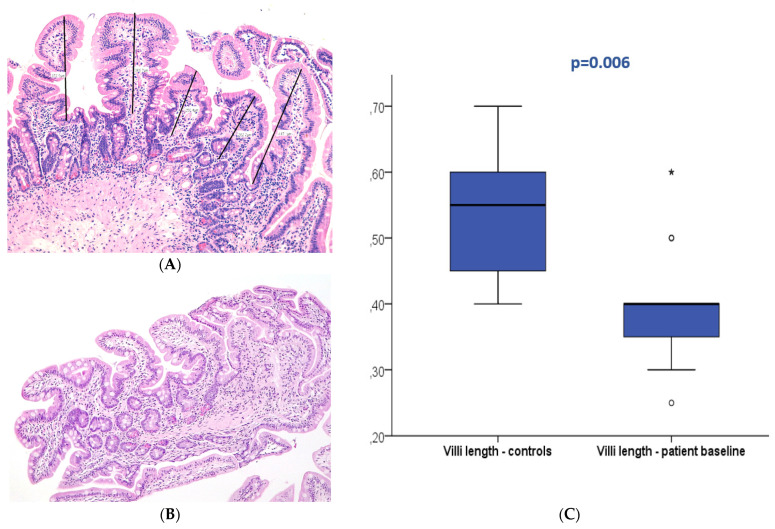
Duodenal mucosa histology stained with hematoxylin and eosin (magnification 100×) exemplifying villi flattening measured at baseline in the patient (**A**) and control groups (**B**). The boxplot (**C**) displays both groups showing significant differences. * and circle are outliers markers.

**Figure 3 nutrients-16-00128-f003:**
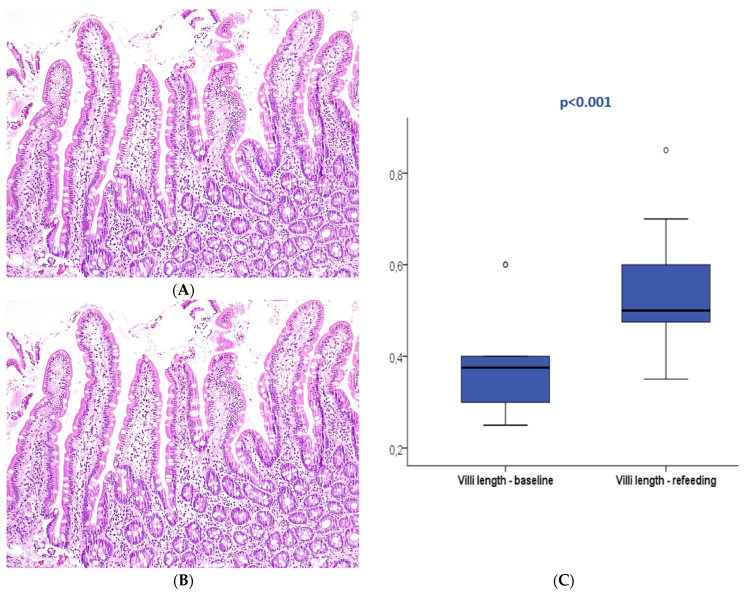
Duodenal mucosa histology stained with hematoxylin and eosin (magnification 100×) describing intestinal villi at T0 (**A**) and after refeeding (**B**). The boxplot (**C**) compares both study periods with significant differences. Circle are outliers markers.

**Figure 4 nutrients-16-00128-f004:**
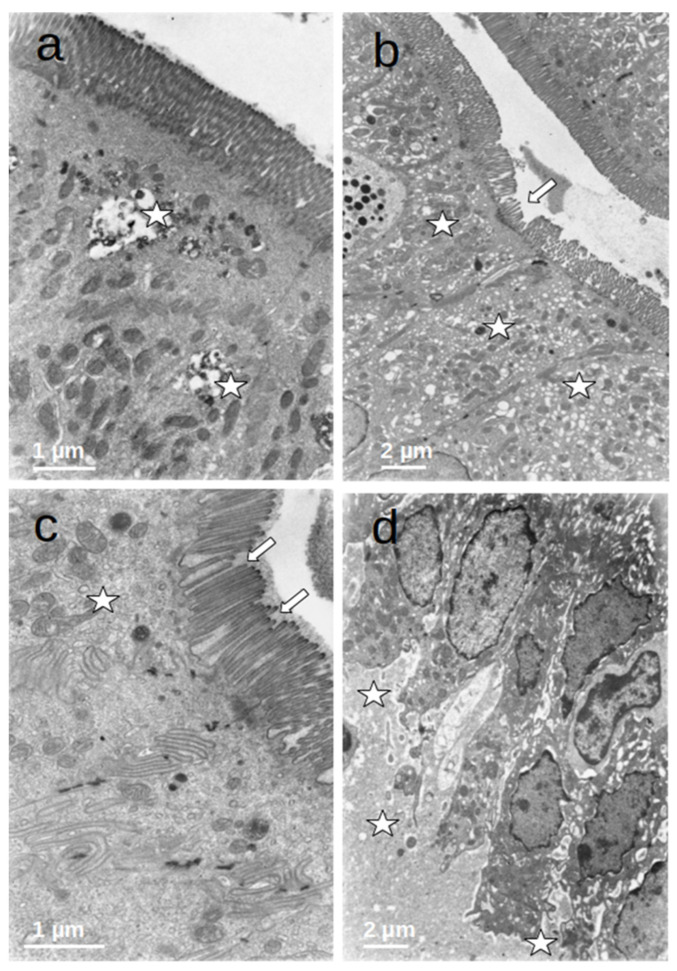
(**a**–**d**) Duodenal mucosa ultrastructure of patients undergoing prolonged fasting. Several degenerative changes can be observed namely: 1. microvilli shortening, bending, and disruption ((**b**,**c**)—arrows); 2. the presence of autophagic vacuoles, dense bodies ((**a**)—asterisks), vesiculation of the smooth endoplasmatic reticulum ((**b**)—asterisks), and heterogeneous mitochondria ((**c**)—asterisk); and 3. dilatation of intercellular spaces that were filled with medium-density granular material ((**d**)—asterisks).

**Figure 5 nutrients-16-00128-f005:**
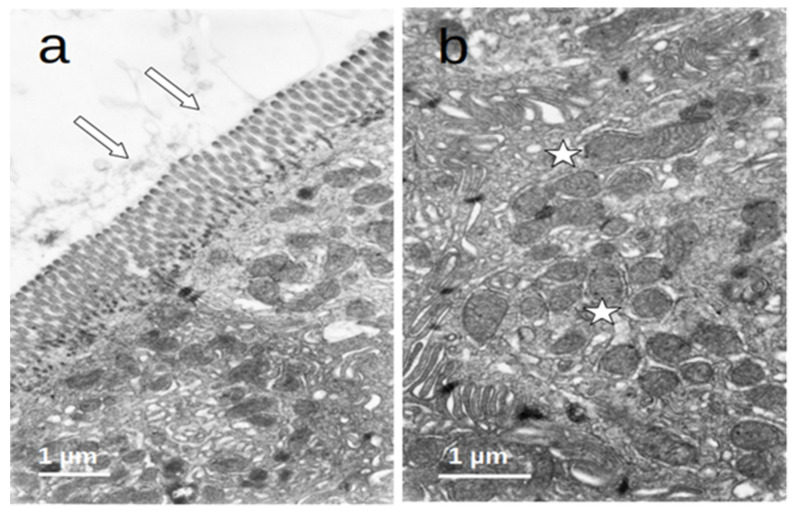
(**a**,**b**) Duodenal mucosa ultrastructure of the controls. The cells display a normal architecture: (**a**) microvilli (arrows) show a regular structure and (**b**) the cytoplasm (asterisks) has a conserved ultrastructure with normal organelles.

**Figure 6 nutrients-16-00128-f006:**
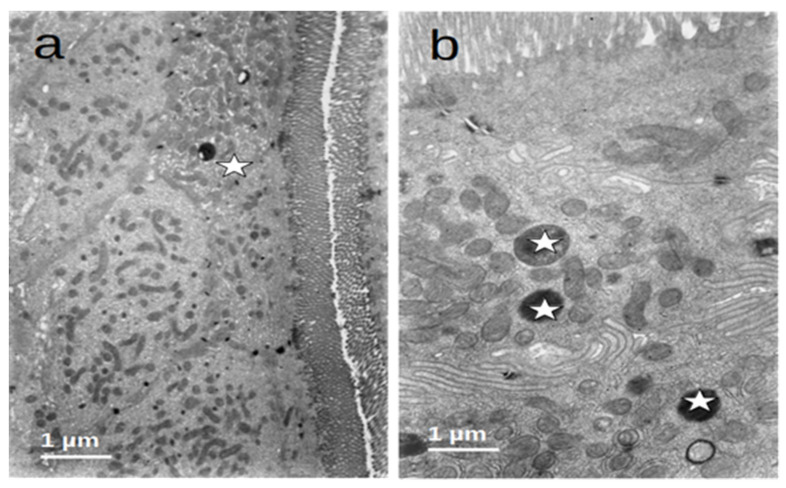
(**a**,**b**) Duodenal mucosa ultrastructure of patients after enteral nutrition. Normal structure of microvilli and cytoplasm ((**a**)—asterisk) and the presence of residual bodies probably resulting from the autophagic activity ((**b**)—asterisks).

**Table 1 nutrients-16-00128-t001:** Baseline characteristics of the patients.

Patient Characteristics—Count (%)		
**Age**	M = 67.1 ± 13.5 years	
**Gender**	Male: 16 Female: 14	(53.3) (46.7)
**Indication for PEG**	**Amyotrophic lateral sclerosis**: 27 **Post-stroke**: 2 **Esophageal dysmotility:** 1	(90) (6.7) (3.3)
**Body Mass Index**	**Low**: 12 **Normal:** 18	(40) (60)
**Albumin**	**Low**: 1 **Normal**: 29	(3.3) (96.7)
**Transferrin**	**Low**: 9 **Normal**: 21	(30) (70)
**Total Cholesterol**	**Low**: 8 **Normal**: 22	(26.7) (73.3)
**Duodenal Histology**	**Normal**: 27 **Marsh 3A**: 2 **Marsh 3B**: 1	(90) (6.7) (3.3)
**Duodenal Villi Length**	M = 0.4 ± 0.09 mm	
	**<0.5 mm:** 22 **≥0.5 mm:** 8	(73.3) (26.7)
**Duodenal Ultrastructure**	Shortening, bending, and disruption of microvilli Autophagic vacuoles Dilation and vesiculation of the ER Dilated intercellular spaces Basal membrane detachment	

**Table 2 nutrients-16-00128-t002:** Optical microscopy analysis and follow-up of the 30 included patients. Villi length increased from T0 to T1 in all patients who completed follow-up, and an improvement in duodenal mucosa was observed in the few patients who displayed atrophy at baseline. Villi length could not be assessed in two patients in T0 due to severe artifact after mucosa sampling and processing.

Patient	Histology (T0)	Villi Length (T0)	Villi Length (T1)	Histology (T1)
**1**	Normal	0.6	0.6	Normal
**2**	Marsh IIIB	0.25	0.4	Marsh IIIA
**3**	Normal	0.6	0.85	Normal
**4**	Normal	0.4	0.6	Normal
**5**	Marsh IIIA	0.3	0.35	Normal
**6**	Normal	0.3	0.5	Normal
**7**	Normal	0.4	0.45	Normal
**8**	Normal	0.35	0.5	Normal
**9**	Normal	0.4	0.7	Normal
**10**	Normal	0.3	0.6	Normal
**11**	Normal	0.4	0.5	Normal
**12**	Marsh IIIA	0.3	0.5	Normal
**13**	Normal	0.35	0.5	Normal
**14**	Normal	Artefact	0.5	Normal
**15**	Normal	0.4	PEG removed	
**16**	Normal	0.5	Deceased	
**17**	Normal	0.4	Deceased	
**18**	Normal	0.6	Deceased	
**19**	Normal	0.4	Deceased	
**20**	Normal	0.4	Deceased	
**21**	Normal	0.3	Deceased	
**22**	Normal	0.4	Deceased	
**23**	Normal	0.4	Deceased	
**24**	Normal	Artefact	Deceased	
**25**	Normal	0.35	Lost for follow-up	
**26**	Normal	0.4	Lost for follow-up	
**27**	Normal	0.4	Lost for follow-up	
**28**	Normal	0.5	Lost for follow-up	
**29**	Normal	0.4	Lost for follow-up	
**30**	Normal	0.4	Lost for follow-up	

## Data Availability

The data presented in this study are available on request from the corresponding author.

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
