# Peer review of "Prolonged Fasting Induces Histological and Ultrastructural Changes in the Intestinal Mucosa That May Reduce Absorption and Revert after Enteral Refeeding"

_nutrients, 2023, doi:10.3390/nu16010128_

Round 1

Reviewer 1 Report

Comments and Suggestions for Authors

This is a very very interesting and well planned study. Good presentation and maths, even if the numbers are small. On the other hand I noted several english language pitfalls, but more importantly an usupported by the results conclusion: the villous atrophy malabsorption, given the fact that starvation leads to villous atrophy, cannot be the cause of the low refeeding syndrome incidence in PEG fed patients, since it would prevent the refeeding syndrome in all patients without discrimination. It is correct to get the electrolytes levels before and  after placing a PEG, in order to evaluate the presence and the severity of a potential refeeding snd, but these should be reported.    

Comments on the Quality of English Language

There are several mistakes needing corrections.  

Author Response

Reviewer 1:

  1. This is a very interesting and well planned study. Good presentation and maths, even if the numbers are small. On the other hand I noted several english language pitfalls, but more importantly an usupported by the results conclusion: the villous atrophy malabsorption, given the fact that starvation leads to villous atrophy, cannot be the cause of the low refeeding syndrome incidence in PEG fed patients, since it would prevent the refeeding syndrome in all patients without discrimination. It is correct to get the electrolytes levels before and after placing a PEG, in order to evaluate the presence and the severity of a potential refeeding syndrome, but these should be reported.  

  1. Thank you for your comment. The English language was reviewed and improved in all manuscript.

Regarding the conclusion of villous atrophy could induce malabsorption and this may explain the low incidence of Refeeding Syndrome (RS) in PEG-fed patients, actually it was an indirect conclusion. As it was cited in the references, a previous study of our group have retrospectively estimated the low prevalence of RS in PEG-fed patients based on the evolution of serum electrolytes in this population (Nunes G, Brito M, Patita M, et al. Hypophosphatemia before endoscopic gastrostomy predicts higher mortality during the first week and first month post-gastrostomy a risk marker of refeeding syndrome in gastrostomy-fed patients. Nutr Hosp. 2019;36(2):247–252). We agree that this conclusion could be generalized to all malnourished patients submitted to enteral refeeding (through oral or tube feeding), nevertheless our study only have included PEG-fed patients followed in the Artificial Nutrition Outpatient Clinic in order to facilitate the access to the duodenal mucosa and avoid ethical concerns. Based on our sample and study methodology, as well as our previous clinical experience, we can just make firm conclusions regarding RS in PEG-fed patients.

Serum electrolytes were assessed before PEG and PEG-feeding was only started after correction of some abnormal values as it was advised in the guidelines. As it was referred in the results, serum electrolytes were normal at the time of PEG in all patients and did not change significantly after refeeding – the authors have reinforced this in the reviewed manuscript.

Reviewer 2 Report

Comments and Suggestions for Authors

Nunes et al. report on an interesting and relevant, novel observation concerning histological and ultrastructural changes of the intestinal mucosa in severely malnourished (ALS) patients before and 3-6 months after enteral nutritional support via PEG.

 Major items: 

1)    The observation by itself is relevant. The assumption made by the authors that the occurrence of refeeding syndrome may be prevented due to these histological/ultrastructural changes is not substantiated, no single piece of evidence is provided. My advice is to only mention this item as assumption in the discussion and leave it out of the aims (see abstract) and conclusions. 

2)    The included patients with malnutrition were in need for nutritional support. Usually at first enteral nutrition is offered by tube feeding, thereafter a PEG is placed. Perhaps due to the chronic character of ALS, PEG was chosen as first step. This should be mentioned. 

3)    Over 90% of the malnourished patients.  was diagnosed with ALS. ALS has been associated with intestinal disorders: the enteric nervous system may become affected, motor neuron disorders occur, permeability is disturbed and dysbiosis may occur. This should be mentioned. 

4)    the point mentioned by the authors as shortcoming is essential: functional tests for absorption (for instance: iron absorption test ) for permeability ( combined sugar absorption tests) and for functional enterocyte mass (citrulline) for enterocyte ischaemia or damage ( FABP) etc. have not been performed. 

Introduction: informative, aims have been formulated in a correct way (RS not mentioned) correct

Methods: observational study, pre and post PEG feeding data ( histology, EM of duodenum ) in ALS patients. 

Results: data on functional tests are lacking. Is determination of plasma citrulline still an option? (blood samples stored in freezer?) Tables and figures are informative and essential. 

Discussion: relevant. The statement in the conclusion that “impaired absorption may be responsible for a low prevalence of refeeding syndrome” should be skipped. It is an interesting assumption but no single proof of evidence is provided.

Author Response

Nunes et al. report on an interesting and relevant, novel observation concerning histological and ultrastructural changes of the intestinal mucosa in severely malnourished (ALS) patients before and 3-6 months after enteral nutritional support via PEG.

 Major items: 

1)    The observation by itself is relevant. The assumption made by the authors that the occurrence of refeeding syndrome may be prevented due to these histological/ultrastructural changes is not substantiated, no single piece of evidence is provided. My advice is to only mention this item as assumption in the discussion and leave it out of the aims (see abstract) and conclusions. 

The manuscript was changed according with your suggestion.

2)    The included patients with malnutrition were in need for nutritional support. Usually at first enteral nutrition is offered by tube feeding, thereafter a PEG is placed. Perhaps due to the chronic character of ALS, PEG was chosen as first step. This should be mentioned. 

Actually, in most patients enteral nutrition starts with oral nutrition supplements followed by tube feeding when it is not sufficient. When tube feeding is indicated for long term enteral nutrition (more than 3-4 weeks) PEG is advised. In most ALS patients, malnutrition develops when dysphagia appeared. In this setting, the disease rapidly progresses and long-term enteral feeding is the rule. PEG placement should not be postponed since respiratory failure is also common in ALS patients which increased mortality associated with the PEG procedure. In our sample, all patients starts tube feeding by PEG – this information was added to the manuscript.

3)    Over 90% of the malnourished patients was diagnosed with ALS. ALS has been associated with intestinal disorders: the enteric nervous system may become affected, motor neuron disorders occur, permeability is disturbed and dysbiosis may occur. This should be mentioned. 

This information was added to the discussion as you have suggested.

4)    The point mentioned by the authors as shortcoming is essential: functional tests for absorption (for instance: iron absorption test for permeability (combined sugar absorption tests) and for functional enterocyte mass (citrulline) for enterocyte ischaemia or damage (FABP) etc. have not been performed. 

This study was a morphologic study and aims only include the description of the histologic and ultrastructural changes of the intestinal mucosa. In fact, there is no gold standard method to study absorption in this clinical setting. The functional tests that you mention were not widely available in our institution and even in our country. An immunohistochemistry-based analysis of intestinal mucosa to study the expression of digestive hormones is also planned by our group to overcome this bias and reinforce the obtained results. Metabolomics may also help to assess the impact of fasting and subsequent refeeding in absorptive mechanisms. Also, further studies with similar patient population are planned aiming to define possible serum biomarkers involved in refeeding syndrome and its impact in nutritional and clinical prognosis. This data was added to the discussion.

Introduction: informative, aims have been formulated in a correct way (RS not mentioned) correct.

Methods: observational study, pre and post PEG feeding data (histology, EM of duodenum) in ALS patients. 

Results: data on functional tests are lacking. Is determination of plasma citrulline still an option? (blood samples stored in freezer?) Tables and figures are informative and essential. 

Discussion: relevant. The statement in the conclusion that “impaired absorption may be responsible for a low prevalence of refeeding syndrome” should be skipped. It is an interesting assumption but no single proof of evidence is provided.

Those topics were changed in the manuscript according with your suggestions with exception for the functional tests which were not performed. Plasma citrulline cannot be determinated at the moment and this serum analysis is not widely available in our institution. Further studies with a different design are planned to overcome this limitation.

Reviewer 3 Report

Comments and Suggestions for Authors

Although some animal model studies have demonstrated that long periods of fasting may induce morphological changes in intestinal mucosa, this paper confirms that this can and does occur in humans. In a small but clinically relevant, prospectively recruited, case series 30 patients (most with neurological dysphagia) referred for PEG had oral ingestion below 50% of energy daily needs for a minimum period of one month and/or baseline malnutrition according to ESPEN guideline. Additionally, 10 control patients investigated for anemia and without pathology in the upper GI tract were also assessed. Appropriate clinical assessment and pathological investigations were performed.

Of the patients, at recruitment, 3/30 (10%) had intestinal mucosa atrophy and 22/30 had short villi, presumably induced by malnutrition. The authors propose that median villi length <0.5mm could be a surrogate marker of subtle mucosal dysfunction that may lead to impaired absorption. In support of this hypothesis, these changes had recovered in all but one following 3-6 months of PEG feeding (14 patients were studied on both occasions)

Comment and critique

This is a small but well designed and performed case series that provides interesting insight into the response of human intestinal mucosa to malnutrition. As is typical in clinical studies the group was heterogenous and follow up was incomplete. Nevertheless interesting longitudinal observations were obtained. Results are clearly presented and the authors (cautious) conclusions are supported by the findings.

The introduction is clear. ; however, the focus is on refeeding syndrome... a condition that did not occur / is not studied in this prospective case series. The authors should rewrite the introduction to focus on the main topic, specifically the response of intestinal mucosa in humans to malnutrition and the recovery of these abnormalities with enteral feeding.

Author Response

Thank you very much for your comments. 

The authors have performed some changes in the manuscript to avoid the indirect conclusion of low prevalence of refeeding syndrome due to duodenal mucosa histological and US changes, especially in the abstract as it was already suggested by another reviewer. 

In the introduction,  the authors would like to have discussed the response of intestinal mucosa in humans to malnutrition and the recovery of these abnormalities with enteral feeding nevertheless this data is not clearly available in the literature as it was referred for animal models - this also demonstrate the innovative concept of this study.

We just maintain the paragraph summarizing RS to improve the length of the manuscript - it was suggested by the editor.